

# UltraTimTrack: a Kalman-filter-based algorithm to track muscle fascicles in ultrasound image sequences

Tim J. van der Zee[1,2,3,4], Paolo Tecchio[3], Daniel Hahn[3,5] and Brent J. Raiteri[3,5]

[1] Biomedical Engineering Graduate Program, University of Calgary, Calgary, Canada
[2] Faculty of Kinesiology, University of Calgary, Calgary, Canada
[3] Department of Human Movement Science, Ruhr University Bochum, Bochum, Germany
[4] Department of Movement Sciences, Katholieke Universiteit Leuven, Leuven, Belgium
[5] School of Human Movement and Nutrition Sciences, University of Queensland, Brisbane, Australia

Corresponding author
Brent J. Raiteri, brent.raiteri@ruhr-uni-bochum.de, brent.raiteri@rub.de

## ABSTRACT

**Background**. Brightness-mode (B-mode) ultrasound is a valuable tool to non-invasively image skeletal muscle architectural changes during movement, but automatically tracking muscle fascicles remains a major challenge. Existing fascicle tracking algorithms either require time-consuming drift corrections or yield noisy estimates that require post-processing. We therefore aimed to develop an algorithm that tracks fascicles without drift and with low noise across a range of experimental conditions and image acquisition settings.

**Methods**. We applied a Kalman filter to combine fascicle length and fascicle angle estimates from existing and openly-available UltraTrack and TimTrack algorithms into a hybrid algorithm called UltraTimTrack. We applied the hybrid algorithm to ultrasound image sequences collected from the human medial gastrocnemius of healthy individuals ($N = 8$, four women), who performed cyclical submaximal plantar flexion contractions or remained at rest during passive ankle joint rotations at given frequencies and amplitudes whilst seated in a dynamometer chair. We quantified the algorithm's tracking accuracy, noise, and drift as the respective mean, cycle-to-cycle variability, and accumulated between-contraction variability in fascicle length and fascicle angle. We expected UltraTimTrack's estimates to be less noisy than TimTrack's estimates and to drift less than UltraTrack's estimates across a range of conditions and image acquisition settings.

**Results**. The proposed algorithm yielded low-noise estimates like UltraTrack and was drift-free like TimTrack across the broad range of conditions we tested. Over 120 cyclical contractions, fascicle length and fascicle angle deviations of UltraTimTrack accumulated to $2.1 \pm 1.3$ mm (mean $\pm$ sd) and $0.8 \pm 0.7$ deg, respectively. This was considerably less than UltraTrack ($67.0 \pm 59.3$ mm, $9.3 \pm 8.6$ deg) and similar to TimTrack ($1.9 \pm 2.2$ mm, $0.9 \pm 1.0$ deg). Average cycle-to-cycle variability of UltraTimTrack was $1.4 \pm 0.4$ mm and $0.6 \pm 0.3$ deg, which was similar to UltraTrack ($1.1 \pm 0.3$ mm, $0.5 \pm 0.1$ deg) and less than TimTrack ($3.5 \pm 1.0$ mm, $1.4 \pm 0.5$ deg). UltraTimTrack was less affected by experimental conditions and image acquisition settings than its parent algorithms. It also yielded similar or lower root-mean-square deviations from manual tracking for previously published image sequences (fascicle length: 2.3–2.6 mm, fascicle angle: 0.8–0.9 deg) compared with a recently-proposed

---

hybrid algorithm (4.7 mm, 0.9 deg), and the recently-proposed DL_Track algorithm (3.8 mm, 3.9 deg). Furthermore, UltraTimTrack's processing time (0.2 s per image) was at least five times shorter than that of these recently-proposed algorithms.
**Conclusion**. We developed a Kalman-filter-based algorithm to improve fascicle tracking from B-mode ultrasound image sequences. The proposed algorithm provides low-noise, drift-free estimates of muscle architectural changes that may better inform muscle function interpretations.

## INTRODUCTION

Brightness-mode (B-mode) ultrasonography, or ultrasound, is a non-invasive method for looking under the skin to image the human body's tissues, which can be applied to study skeletal muscle function during movement (*Rutherford & Jones, 1992*; *Fukunaga et al., 2001*). During passive movements and active muscle contraction, ultrasound can be used to visualize the connective tissue around muscle fascicles—bundles of muscle fibers—and their changes in length and orientation. Fascicle length and angle affect a muscle's force potential (*Azizi, Brainerd & Roberts, 2008*; *Bohm et al., 2019*) and metabolic energy expenditure (*Joumaa & Herzog, 2013*; *van der Zee, Lemaire & van Soest, 2019*; *van der Zee & Kuo, 2021*; *Beck et al., 2022*), thereby affecting movement performance and economy (*Fletcher & MacIntosh, 2017*; *Swinnen et al., 2021*; *Schwaner et al., 2024*). Consequently, quantifying fascicle length and angle changes is important for improving our understanding of *in vivo* muscle-tendon function (*Cronin & Lichtwark, 2013*). Recent years have hence seen an increase in algorithms to automate muscle ultrasound image analysis and fascicle tracking, which had previously been limited by the laborious and subjective nature of manual labelling (*van Hooren, Teratsias & Hodson-Tole, 2020*). However, existing fascicle tracking algorithms are still error prone and have limitations that prevent complete automatization (*Ritsche et al., 2024*). It would thus be helpful to develop an algorithm that automatically and robustly tracks muscle architectural changes from ultrasound image sequences to more quickly and easily interpret muscle-tendon function during movement.

Optical flow is a commonly used method to track muscle fascicles in a 'semi-automated' manner (*Gillett, Barrett & Lichtwark, 2013*). Optical-flow-based algorithms such as UltraTrack (*Farris & Lichtwark, 2016*) make a best guess (*i.e.,* least-squares approximation) of the apparent motion between consecutive ultrasound images to track muscle fascicles (*Farris & Lichtwark, 2016*; *Drazan, Hullfish & Baxter, 2019*). This method is relatively insensitive to the speckle noise present in ultrasound images because comparing two consecutive frames effectively removes common noise. However, small tracking errors can occur in each frame, because of image brightness changes or variable local pixel motion. Integrated over many frames, these tracking errors accumulate, causing fascicle length

and fascicle angle estimates to 'drift' away from their original values (*Magana-Salgado et al., 2023*). To correct for this drift, additional information (*e.g.*, timing of external forces) is required. Additionally, if there is substantial motion between consecutive frames, manual corrections are required to avoid underestimation of fascicle length and fascicle angle changes. Fascicles also need to be defined before tracking, which is typically done *via* manual labelling. Consequently, while the noise insensitivity of optical-flow-based methods is an important advantage for tracking small fascicle displacements (*e.g.*, those that occur during healthy postural sway (*Day et al., 2017*)) over a few frames, drift sensitivity and manual labelling requirements limit their accuracy, objectivity, repeatability, and time effectiveness in other conditions (*e.g.*, large fascicle displacements during locomotion).

Unlike optical-flow-based methods, line-detection-based and artificial-intelligence (AI)-based methods attempt to automatically identify line segments in individual muscle ultrasound images. Line-detection-based algorithms such as TimTrack (*van der Zee & Kuo, 2022*) analyse each ultrasound image independently, making them insensitive to drift (*Zhou & Zheng, 2008*; *Rana, Hamarneh & Wakeling, 2009*; *Zhou et al., 2012*; *Zhou, Chan & Zheng, 2015*; *Marzilger et al., 2018*; *Ryan et al., 2019*; *Seynnes & Cronin, 2020*; *van der Zee & Kuo, 2022*). However, because similarities and differences between consecutive images are not considered, these methods are sensitive to the speckle noise present in ultrasound images. Similarly, recently-proposed AI-based algorithms (*Bao et al., 2023*; *Ritsche et al., 2024*; *Yuan et al., 2024*) also analyse each image independently, yielding noisier estimates compared with optical-flow-based methods (*Ritsche et al., 2024*). Both line-detection-based and AI-based algorithms thus yield relatively noisy estimates of muscle architectural changes during movement that require low-pass filtering within a trial or averaging over multiple trials. Consequently, while line-detection-based and AI-based methods achieve high degrees of automation favoring objectivity, repeatability, and time effectiveness, their sensitivity to noise reduces tracking accuracy and usability. This may explain why recent biomechanical studies (*e.g.*, *Swinnen et al., 2024*; *van Hooren et al., 2024*; *Beck, Schroeder & Sawicki, 2024*; *Raiteri, Lauret & Hahn, 2024*; *Farris et al., 2024*) still use UltraTrack for tracking muscle architectural changes during movement, despite its manual labelling requirements and drift sensitivity.

Here, we propose to leverage the key advantages and overcome the main limitations of existing fascicle tracking algorithms by combining existing techniques into a new Kalman-filter-based fascicle tracking algorithm. We chose Kalman filtering (*Kalman, 1960*) because it is a popular sensor-fusion method that has been successfully employed to correct drift (*e.g.*, *Lee & Jung, 2009*) and reduce noise (*e.g.*, *Alfian, Ma'arif & Sunardi, 2021*) in robotics and other fields. The proposed algorithm combines estimates from a noise-insensitive, but drift-sensitive, optical flow method (*i.e.*, Kanade-Lucas-Tomasi optical flow (*Lucas & Kanade, 1981*; *Shi & Tomasi, 1994*)) with drift-free, but noise-sensitive, line-detection methods (*i.e.*, object detection and Hough-transform methods (*Duda & Hart, 1972*)) to yield improved estimates of muscle architectural changes (Fig. 1). We re-used and modified open-source code from existing UltraTrack (*Farris & Lichtwark, 2016*) and TimTrack (*van der Zee & Kuo, 2022*) algorithms, and therefore coined the proposed algorithm 'UltraTimTrack'. The proposed algorithm and its parent algorithms

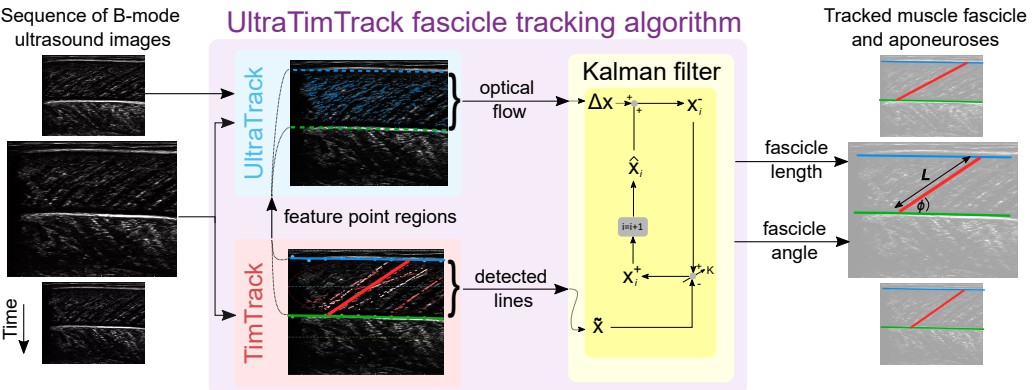

**Figure 1** **Overview of the UltraTimTrack algorithm.** Left panel: Sequence of Brightness-mode (B-mode) ultrasound images. Middle panel: UltraTimTrack algorithm consists of an UltraTrack module, a TimTrack module, and a Kalman filter. First, the TimTrack module detects lines in the image corresponding to either aponeuroses or fascicles. The UltraTrack module then defines feature points near TimTrack's detected lines and calculates the optical flow that captures the displacement of these points between the current and previous image. Optical flow and detected lines are input to the Kalman filter. In each $i$-th iteration, the change in estimated state is obtained from optical flow to yield an *a-priori* estimate, which is updated with detected lines in proportion to a Kalman gain K to yield an a-posteriori estimate. State estimates yield estimates of the tracked fascicle and its length L and angle with the deep aponeurosis $\varphi$. Right panel: Tracked muscle fascicle and aponeuroses displayed on the original images.

were tested on various B-mode ultrasound image sequences collected from the left-sided medial gastrocnemius muscle of healthy human participants. The proposed algorithm was also applied to two previously collected ultrasound image sequences from the human tibialis anterior and medial gastrocnemius muscles and compared with two recently-proposed algorithms. One of these algorithms considers a hybrid method that combines optical-flow-based and line-detection-based algorithms (*Verheul & Yeo, 2023*), but lacks a Kalman filter (here referred to as HybridTrack) and the other is the AI-based DL_Track algorithm (*Ritsche et al., 2024*). We expected UltaTimTrack to be noise-insensitive like optical-flow-based algorithms, and to be drift-free like line-detection-based and AI-based algorithms. Portions of this manuscript were previously published as part of a preprint (https://doi.org/10.1101/2024.08.07.607010).

# METHODS

## Synopsis

We developed a Kalman-filter-based fascicle tracking algorithm that combines tracking estimates from existing and openly-available algorithms to yield improved estimates of muscle fascicle length and fascicle angle changes during movement. The proposed UltraTimTrack algorithm was evaluated using ultrasound image sequences collected from the left-sided medial gastrocnemius muscle of healthy young adults during cyclical submaximal voluntary fixed-end plantar flexion contractions at various frequencies with varying activation levels, as well as during passive ankle rotations at various angular

velocities. We first describe the algorithm before discussing the experimental methods, expected outcomes and statistics.

## Proposed algorithm

To improve the accuracy of the proposed UltraTimTrack algorithm, we first modified one of its parent algorithms as discussed immediately below. We then describe the Kalman filter, followed by the graphical user interface.

### *Parent algorithms of UltraTimTrack*

The proposed algorithm has UltraTrack and TimTrack modules (Fig. 1), which are based on the corresponding original algorithms. However, notable modifications were made to UltraTrack to improve its fascicle tracking performance. Firstly, the UltraTrack module of UltraTimTrack employs Kanade-Lucas-Tomasi optical flow (*Shi & Tomasi, 1994*), while the original UltraTrack algorithm (*Farris & Lichtwark, 2016*) employs Lucas-Kanade optical flow (*Lucas & Kanade, 1981*). The main difference is that Kanade-Lucas-Tomasi only tracks 'good feature points' that can be tracked well (in our case, corner points), which improves tracking accuracy while reducing computational cost. The Kanade-Lucas-Tomasi method has been used in more recent optical-flow-based fascicle tracking algorithms (*e.g.*, *Drazan, Hullfish & Baxter, 2019*), including later and improved, but non-peer-reviewed, versions of UltraTrack (*Bakenecker et al., 2022*; *Raiteri, Lauret & Hahn, 2024*; *Tecchio, Raiteri & Hahn, 2024*). Similar to these versions, MATLAB's (MathWorks, Inc., Natick, MA, USA) built-in detectMinEigenFeatures, the PointTracker and estimateGeometricTransform2D functions were used to detect feature points and compute optical flow with an affine transformation type, respectively. Optical flow parameters were set to values used in the most recent version of UltraTrack (*Raiteri, Lauret & Hahn, 2024*). Secondly, unlike any previous UltraTrack version, the UltraTrack module of UltraTimTrack separately computes optical flow for fascicles and aponeuroses. It leverages TimTrack's detection of fascicle and aponeurosis locations to more precisely identify feature points that are specific to these structures. More specifically, it uses the TimTrack module to identify superficial- and deep-aponeurosis locations within user-specified regions (default: 5-30% and 30-70% of vertical image range, with 0% referring to the top of the cropped ultrasound image). With the region of interest (ROI) type (Fig. 2) set to 'Hough-local' (default), feature points are then selected in distinct, local fascicle regions that are based on TimTrack's detected fascicle locations. These fascicle regions are rectangles of fixed width (default: 10 pixels), centered around the locations of the most frequently occurring lines (default: 10) detected by TimTrack. Alternatively, ROI type can be set to 'Hough-global' to allow feature points to be detected within the whole ROI between aponeuroses. In both cases, regions outside the middle portion (default: 80%) of the overall region between TimTrack's aponeuroses are excluded to avoid detecting feature points on aponeuroses, like in an existing algorithm (*Drazan, Hullfish & Baxter, 2019*). A fixed number of feature points is selected within the remaining fascicle region (default: 300) using MATLAB's built-in selectStrongest function.

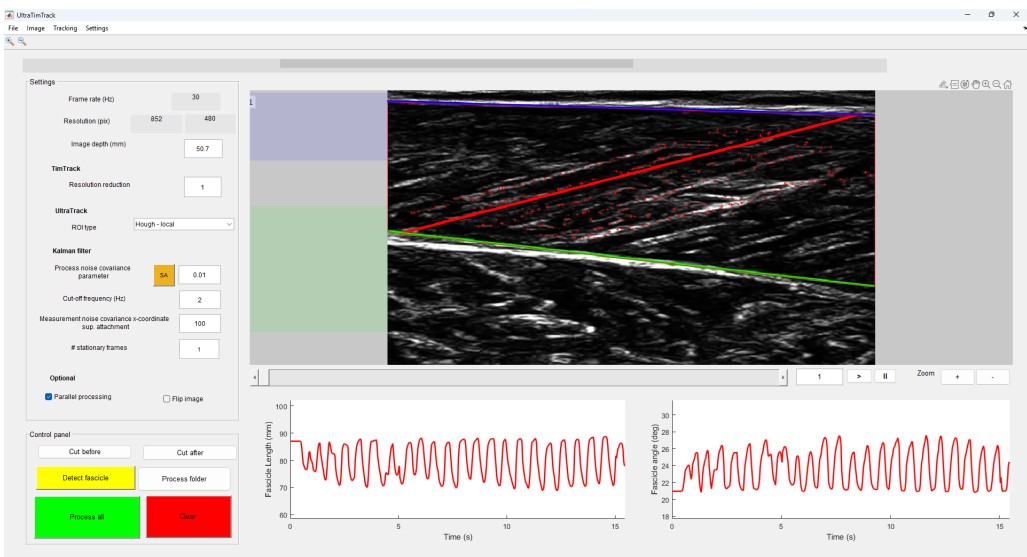

**Figure 2** **Graphical user interface of the UltraTimTrack algorithm.** Users can inspect the tracked fascicle in the current ultrasound image, along with the fascicle length and fascicle angle estimates from the entire sequence. The analyzed image indicates the tracked fascicle (red line), superficial aponeurosis (blue line), deep aponeurosis (green line), and their respective feature points (red plus symbols). The aponeuroses are detected within specified regions (shaded blue and green areas to the left of the image). Users can specify the settings (top left column) and run the algorithm (bottom left column) for either the entire image sequence (Process all) or just the current frame (Detect fascicle). Settings are explained in the main text.

### Kalman filter

We now briefly describe the Kalman filter and then discuss how it was applied to track muscle fascicles. The reader is referred to other literature for a more detailed description of Kalman filters (*Kalman, 1960*), summarized in Article S1.

A Kalman filter considers the state (vector) $z$ of a dynamical system at time $i$. It assumes that the current state (vector) $z_i$ depends on state-transition model $A$, control-input model $B$, the previous state (vector) $z_{i-1}$, the current input $u_i$, and the current process noise $w_i$:

$$z_i = Az_{i-1} + Bu_i + w_i. \tag{1}$$

The process noise $w_i$ is unknown, and assumed to be drawn from a zero-mean normal distribution with variance $Q_i$. Because the process noise $w_i$ is unknown, state $z_i$ cannot be computed directly. Instead, the Kalman filter can predict state $z_i$, and update this prediction using a measurement $\tilde{z}_i$ to yield an optimal state estimate $\hat{z}_i$.

In the proposed implementation, state $z_i$ is obtained through time-integrating a rate-based input $u_i$. Thus, the modelled system is an integrator, with the state-transition model $A$ equal to unity and the control-input model $B$ equal to the time difference between $i$ and $i-1$.

*States and measurements* The proposed fascicle tracking algorithm performs Kalman filtering on both aponeuroses and fascicles. For aponeuroses, the states are the vertical

($y$) positions of the superficial ($S$) and deep ($D$) aponeuroses at the left ($L$) and right ($R$) image boundaries (*i.e.*, $S_{L,y}$, $S_{R,y}$, $D_{L,y}$, $D_{R,y}$). For the fascicle, the states are the fascicle orientation ($\alpha$, with respect to the horizontal) and the horizontal ($x$) position of a point $p$ on the fascicle ($p_x$). For both aponeuroses and the fascicle, UltraTrack's optical flow is treated as input $u$, and TimTrack's line detection is treated as the measurement $\bar{z}$. An *a priori* state estimate is obtained by applying UltraTrack to the previous state estimate (equivalent to $Az_{i-1} + Bu$), which is then updated using TimTrack to yield an *a posteriori* state estimate. More specifically, aponeurosis locations and the fascicle orientation are updated with TimTrack's aponeurosis location estimates and fascicle orientation estimate, respectively. The fascicle point estimate is updated using a constant that is equal to that defined in the first frame. Finally, a Rauch-Tung-Striebel smoother (*Rauch, Striebel & Tung, 1965*) is applied to smooth the state estimates (see Article S1 for details).

*Fascicle and aponeurosis locations*  The vertical positions of the superficial aponeurosis, deep aponeurosis, and fascicle can be computed from the states and expressed as functions of horizontal ($x$) position, yielding $S(x)$, $D(x)$ and $F(x)$, respectively:

$$S(x) = S_{L,y} + \frac{S_{R,y} - S_{L,y}}{r}x \tag{2}$$

$$D(x) = D_{L,y} + \frac{D_{R,y} - D_{L,y}}{r}x \tag{3}$$

$$F(x) = p_y + \tan(\alpha)(x - p_x). \tag{4}$$

Here, $r$ is the image width and $p_y$ is the vertical position of point $p$ (Table 1 for symbol definitions).

*Fascicle tracking estimates*  Two main fascicle tracking estimates are derived from the proposed method: (1) fascicle angle $\phi$ and (2) fascicle length $L$. Here, we define fascicle angle $\phi$ as the angle of the fascicle relative to the straight-line projection of the deep aponeurosis in the ultrasound image. First, the superficial and deep attachment points $[s_x, s_y]$ and $[d_x, d_y]$ are found by solving $F(x) = S(x)$ and $F(x) = D(x)$, respectively. Next, these attachment points are combined with the aponeurosis locations to compute fascicle angle $\phi$ (*i.e.*, with respect to the deep aponeurosis) and fascicle length $L$:

$$L = \sqrt{(s_y - d_y)^2 + (s_x - d_x)^2} \tag{5}$$

$$\phi = \text{atan}\left(\frac{s_y - d_y}{s_x - d_x}\right) - \text{atan}\left(\frac{D_{R,y} - D_{L,y}}{r}\right). \tag{6}$$

For UltraTrack, the second term of Eq. (6) was set to 0 because UltraTrack requires additional inputs to track aponeuroses.

*Noise estimation*  To determine the Kalman gain (see Article S1), the process noise and measurement noise covariances need to be estimated. The measurement noise covariance reflects the uncertainty in TimTrack's line detection, which is affected by image (speckle) noise. We estimated the covariance of this noise by taking the variance of the high-pass filtered TimTrack aponeurosis locations and fascicle orientations, using a second-order

**Table 1  Variables of the UltraTimTrack algorithm.**

| Symbol | Type | Meaning |
|---|---|---|
| $S_{L,y}$ | State | Vertical position of the superficial aponeurosis at the left image boundary |
| $S_{R,y}$ | State | Vertical position of the superficial aponeurosis at the right image boundary |
| $D_{L,y}$ | State | Vertical position of the deep aponeurosis at the left image boundary |
| $D_{R,y}$ | State | Vertical position of the deep aponeurosis at the right image boundary |
| $\alpha$ | State | Fascicle orientation with respect to the horizontal |
| $p_x$ | State | Horizontal position of the fascicle point $p$ |
| $p_y$ | Constant | Vertical position of the fascicle point $p$ |
| $r$ | Constant | Image width |
| $s_x$ | Variable | Horizontal position of the superficial fascicle attachment point |
| $s_y$ | Variable | Vertical position of the superficial fascicle attachment point |
| $d_x$ | Variable | Horizontal position of the deep fascicle attachment point |
| $d_y$ | Variable | Vertical position of the deep fascicle attachment point |
| $L$ | Variable | Fascicle length |
| $\phi$ | Variable | Fascicle angle with respect to the deep aponeurosis |

dual-pass Butterworth filter (default cut-off frequency $f_c$: 2 Hz). The measurement noise covariance of the horizontal position of the fascicle point $p$ reflects how well the initial position value reflects the instantaneous position value. A large covariance allows more movement of this point, but at the cost of more drift (default: 1000 pixels). Considering that optical flow is most accurate for small displacements, the process noise $Q$ was assumed to increase with the square of the optical flow input $u$:

$$Q_i = cu_i^2. \tag{7}$$

The proportionality constant $c$ is unknown, and it is chosen through trial-and-error (default: 0.01). We later report on a sensitivity analysis on the effect of parameter $c$ on tracking accuracy.

*Initial state estimate*  The initial state estimate $\hat{z}_1$ is automatically determined by evaluating the TimTrack module on a user-defined number of stationary initial frames (default: 1). The initial state estimate $\hat{z}_1$ equals the mean TimTrack module estimate over these initial frames. Initial frames must be stationary (*e.g.*, at a steady-state passive or active muscle force during no muscle length change). If no stationary frames are available, the initial state estimate $\hat{z}_1$ should be determined based on the first frame only.

### Graphical user interface

UltraTimTrack has a similar graphical user interface (GUI) as UltraTrack (Fig. 2). It displays the current ultrasound image and the tracked muscle fascicle, alongside the estimated fascicle lengths and fascicle angles from the image sequence. The algorithm can be started using the "Process all" button and reset using the "Clear" button. The "Detect fascicle" button detects a single fascicle from the current image, and the "Process folder" button allows all videos in a specified folder to be processed. Settings include the frame rate and resolution (obtained from the video file), and the image depth to scale pixels

to mm (note that if a TVD file is converted to an MP4 file using the provided function, TVD2ALL.m, this information is stored in a MAT file and automatically loaded along with the video). The image resolution can be optionally decreased with a specified factor ("Resolution reduction") to speed up TimTrack. Under Processing, the user can specify the estimated optical flow process noise covariance parameter $c$ (see "Noise estimation"), the cut-off frequency of the high-pass filtering, the superficial attachment point measurement noise covariance, and the number of stationary frames (see "Initial state estimate"). The "SA" button performs a sensitivity analysis on process noise covariance parameter $c$. Under "Options", the user can also check whether to perform parallel processing to speed up computation, and whether to flip the image about its middle vertical axis. Currently, only forward tracking from the first frame of the image sequence is supported.

## Experimental methods

To test the UltraTimTrack algorithm, ultrasound images from the left-sided medial gastrocnemius muscle of healthy human participants were collected during various active and passive conditions performed within a dynamometer setup.

### Participants

We collected data from healthy young adults ($N = 8$, four women, age $= 26 \pm 3$ years, height $= 174 \pm 6$ cm, weight $= 70 \pm 7$ kg) during ankle plantar flexion contractions of various speeds, various hold durations, and submaximal voluntary intensities. Participants were included based on the following criteria: age 18-45 years, no pre-existing neuromuscular conditions, and no injuries of the lower extremity in the last six months. The latter was determined by a standardised questionnaire that was approved by the Local Ethics Committee. Participants provided written informed consent prior to their voluntary participation in the experiment. The experiment was approved by the Ethics Committee of the Faculty of Sport Science at Ruhr University Bochum (reference: EKS V 19/2022).

### Experimental setup

Participants were seated in a dynamometer chair (IsoMed2000, D&R Ferstl GmbH, Berlin, Germany), with the sole of their left foot flush against the footplate attachment of a motorized dynamometer while their left knee was extended. Their left foot had an attachment above it that was padded to limit forefoot movement. A figure of eight strap around the footplate attachment and participant's lower shank was used to avoid heel lift. The back of the dynamometer chair was reclined to an angle of about 60 deg to avoid stretch of the hamstring muscles.

Net ankle joint torque and crank-arm angle were measured using the dynamometer and sampled at 2 kHz using a 16-bit analog-to-digital converter (Power1401-3, Cambridge Electronic Design Ltd., Cambridge, UK) with a $\pm 5$ V input range, and data collection software (Spike2, 10.10 64-bit version, Cambridge Electronic Design Ltd).

The left-sided medial gastrocnemius muscle was imaged using a linear, flat, 128-element ultrasound transducer (LV8-5N60-A2, Telemed Medical Systems srl, Milan, Italy) that was attached to a PC-based ultrasound beamformer (ArtUS EXT-1H, Telemed Medical Systems, Milan, Italy). Ultrasound images were captured with a $60 \times 50$ mm (width

$\times$ depth) field of view at frame rates ranging from 33 images $s^{-1}$ to 100 images $s^{-1}$. Image collection was synchronized with the collection of torque and crank-arm angle, as previously described (*Raiteri, Lauret & Hahn, 2024*; *Tecchio, Raiteri & Hahn, 2024*).

The activity of the following four lower leg muscles was recorded using surface electromyography (EMG): medial gastrocnemius, lateral gastrocnemius, soleus, and tibialis anterior. EMG signals were recorded using a NeuroLog system (NL905, Digitimer Ltd., Welwyn Garden City, UK). EMG signals were recorded at 2 kHz and synchronised using the previously described analog-to-digital converter and data collection software.

### Experimental procedure

Before performing the experiment, participants were prepared for the EMG and ultrasound measurements. First, we located the muscly belly borders of the left-sided medial gastrocnemius using ultrasound. To improve sound wave transmission, ultrasound gel was applied over the participant's skin. The desired location of the ultrasound transducer over the muscle belly was marked using a red permanent marker. Generally, the ultrasound transducer was placed over the most prominent bulge of the muscle. Next, the EMG electrodes were placed over the skin covering the four muscles of interest according to SENIAM guidelines (*Hermens et al., 2000*). For the medial gastrocnemius, the location was shifted medially or laterally in case it overlapped with the desired location of the ultrasound transducer. For each of the four muscles, an area of about 3-4 $cm^2$ was shaved, exfoliated and disinfected before electrode placement. EMG electrodes were placed with an interelectrode center-to-center distance of 2 cm following SENIAM recommendations (*Hermens et al., 2000*). Next, the ultrasound transducer was secured to the lower leg using a custom-made case and self-adhesive bandage. Then the dynamometer chair was set up, and the dynamometer axis of rotation was aligned to the participant's ankle joint axis of rotation (estimated as the transmalleolar axis) during a 50% MVC contraction at 0 deg plantar flexion (*i.e.,* sole of foot perpendicular to shank). Movement of the dynamometer footplate was restricted to occur within the participant's ankle joint range of motion using both electrical and mechanical stops. After set-up was complete, participants were asked to perform the three distinct experimental conditions described below.

### Experimental conditions

The experiment involved three distinct types of conditions: (1) ~3-s duration maximal voluntary fixed-end contractions (MVCs) of plantar flexion and dorsiflexion at 0 deg plantar flexion; (2) ten prolonged (80 s) bouts of submaximal cyclical fixed-end plantar flexion contractions at different rates at 0 deg plantar flexion, and; (3) passive ankle joint rotations over a 50 deg range of ankle angles at different velocities.

During the MVCs, the participants were verbally encouraged by the experimenter to push or pull with the ball or top of their foot as hard as possible for at least 3 s. After each MVC, participants rested for at least two minutes. For the plantar flexion MVCs, this procedure was repeated until the maximal torque of the subsequent contractions differed by less than 5%, or until five contractions were performed, whichever came first. Following the plantar flexion MVCs, participants performed at least one dorsiflexion MVC.

During the prolonged submaximal cyclical contractions, participants were asked to match their net ankle joint plantar flexion torque to a visually displayed torque target for a duration of 80 s in six sub-conditions. In four of those conditions, the target was to ramp up and down from 0% MVC to 50% MVC, with a 1-sec hold phase at 50% MVC. Three of these four ramp-and-hold conditions had a symmetric target trace that differed in ramp rate (20% MVC $\cdot$s$^{-1}$, 33% MVC $\cdot$s$^{-1}$ and 100% MVC $\cdot$s$^{-1}$). The fourth ramp-and-hold condition involved a different ramp rate up (20% MVC$\cdot$s$^{-1}$) *versus* down (the instruction was to relax as fast as possible). Each ramp-and-hold condition was repeated once to allow for two ultrasound image qualities to be assessed, which resulted in at least eight prolonged contraction bouts. So-called "high" and "low" quality images were obtained with line density software settings of high and standard S, respectively. To reduce the chance of fatigue, participants were given at least two minutes of rest between bouts. The order in which ramp-and-hold conditions were performed was randomized across participants. In the remaining two conditions, the target trace was a sinusoid with a frequency of 1.5 Hz that was not repeated (only a high-image quality was used), which resulted in two prolonged contraction bouts. These two sinusoidal conditions differed in their torque range (0-20% MVC and 10-20% MVC) and their order of completion was randomized.

Following the cyclical contractions, participants were asked to sit still and relax while the experimenter triggered the dynamometer to passively rotate their foot eight times over 50 deg within the participant's ankle joint range of motion. Three passive rotation conditions were performed, each at a different angular velocity in a randomized order: 5 deg$\cdot$s$^{-1}$, 30 deg$\cdot$s$^{-1}$, and 120 deg$\cdot$s$^{-1}$. The corresponding maximum and minimum ankle angles were 41.6 $\pm$ 7.8 deg, 41.6 $\pm$ 7.9 deg and 41.7 $\pm$ 7.8 deg plantar flexion, and 5.9 $\pm$ 6.8 deg, 5.9 $\pm$ 6.7 deg and 6.0 $\pm$ 6.8 deg dorsiflexion (mean $\pm$ sd across participants).

### Data processing

Torque and crank-arm angle were low-pass filtered using second-order dual-pass Butterworth filters ($f_c = 10$ Hz). EMG data were band-pass filtered ($f_c = 30$–500 Hz) (*Hof & van den Berg, 1981*), rectified and then low-pass filtered ($f_c = 10$ Hz), again using a second-order dual-pass Butterworth filter. The filtered EMG data were normalized with respect to the corresponding maximum value obtained during the MVC trials. Finally, filtered torque, angle and EMG data were resampled to a frequency of 100 Hz.

## Expected outcomes
### Comparison with UltraTrack and TimTrack

UltraTimTrack was compared with UltraTrack and TimTrack algorithms as available from open-access GitHub repositories (*i.e.,* https://github.com/brentrat/UltraTrack_v5_3 and https://github.com/timvanderzee/ultrasound-automated-algorithm, respectively). Combining the experimental conditions allowed us to investigate the effect of three factors on fascicle tracking estimates: (1) sequence duration; (2) image-to-image dissimilarity, and; (3) image quality. UltraTrack was expected to be sensitive to sequence duration, as drift accumulates over time. UltraTrack was also expected to be sensitive to image-to-image dissimilarity, because optical flow assumes a small displacement between consecutive images (*Al-Qudah & Yang, 2023*). TimTrack was expected to be primarily affected by

image quality because its sensitivity to noise implies that it may perform better on high-quality images. UltraTimTrack's Kalman filter was expected to exploit UltraTrack's and TimTrack's complementary dependencies. Specifically, UltraTimTrack was expected to be less sensitive to sequence duration and to image-to-image dissimilarity than UltraTrack, and less sensitive to image quality than TimTrack. We used default values for all algorithm parameters, except for a resolution reduction of 2.

*Outcome measures* We estimated tracking accuracy, noise, and drift using a combination of objective measures and comparisons with manual labelling.

*Overall variability.* Fascicle tracking estimates (*i.e.*, fascicle lengths and fascicle angles) from each contraction cycle were resampled as a percentage of contraction cycle (1% spacing). Overall variability was estimated as the mean standard deviation of fascicle tracking estimates across contraction cycles:

$$\text{Overall variability} = \frac{\sum_{j=1}^{M} \sqrt{\frac{\sum_{i=1}^{N} \left(f_{i,j} - \overline{f}_j\right)^2}{N}}}{M}. \tag{8}$$

Here, $f_{i,j}$ is the fascicle tracking estimate (either fascicle length or fascicle angle) of the $i$th contraction cycle and the $j$th resampled point, $\overline{f}_j$ is the cycle-average fascicle tracking estimate of the $j$th resampled point, $N$ is the number of contraction cycles, and $M$ is the number of resampled points (set to 101, for 1% spacing). A portion of this overall variability reflects 'true' variability (*e.g.*, due to torque-matching inaccuracies during cyclical contractions), with the remainder due to tracking errors of the algorithm. The difference in overall variability provides an estimate of algorithm error when comparing algorithms on the same ultrasound image sequences. Overall variability should increase with both drift and noise of each algorithm's estimates.

We used a different variability measure for the passive ankle rotations because the cycle time and number of cycles were variable. We binned each fascicle tracking estimate into 1-degree joint angle bins and computed the standard deviation for each bin. Overall variability was defined as the average of these standard deviations among joint angle bins.

*Cycle-to-cycle variability.* The difference in estimates between two consecutive contraction cycles was determined for each resampled point and for each pair of consecutive cycles, yielding a two-dimensional difference matrix $\delta$ ($M$-by-[$N$-1]). Cycle-to-cycle variability of each algorithm's tracking estimates was quantified using the standard deviation of the cycle-to-cycle difference in fascicle tracking estimates, and averaged over contraction cycles:

$$\text{Cycle-to-cycle variability} = \sum_{i=1}^{N-1} \frac{\sqrt{\frac{\sum_{j=1}^{M}(\delta_{i,j} - \overline{\delta}_i)^2}{M}}}{N-1}. \tag{9}$$

Here, $\delta_{i,j}$ is the difference in fascicle tracking estimates of the $j$th resampled point within the $i$th cycle pair, and $\overline{\delta}_i$ is the average difference across resampled points of the $i$th cycle

pair. Because this measure only compares two consecutive contraction cycles, it does not capture the drift that happens over longer time scales. Instead, it mostly reflects noise and is less affected by drift.

*Cumulative deviation.* Cumulative deviation of the tracking estimates was quantified using the cumulative rectified mean cycle-to-cycle difference of each algorithm's fascicle tracking estimates. Cumulative deviation was evaluated at the final contraction cycle:

$$\text{Cumulative deviation} = \sqrt{\left( \sum_{i=1}^{N-1} \overline{\delta}_i \right)^2}. \tag{10}$$

As this measure only compares the mean difference between contraction cycles, it does not reflect noise over short time scales (*i.e.,* within a cycle). Instead, it mostly reflects drift and is less affected by noise.

### Comparison to HybridTrack and DL_Track

We also compared the tracking accuracy and processing time of UltraTimTrack with that of two recently-proposed ultrasound tracking algorithms: HybridTrack (*Verheul & Yeo, 2023*) and DL_Track (*Ritsche et al., 2024*). Unlike UltraTrack and TimTrack, the more recent HybridTrack and DL_Track algorithms have not yet been compared to manual tracking estimates across a large range of image sequences. It is therefore unclear how their algorithm parameters would need to be adjusted to yield the best tracking accuracy for a given image sequence. We thus chose to test UltraTimTrack against these algorithms on the image sequences of the accompanying publications, for which parameter values had been tuned by their developers. The selected image sequences were from the human tibialis anterior (*Verheul & Yeo, 2023*) and the medial gastrocnemius (*Ritsche et al., 2024*). We compared tracking estimates from all three algorithms with estimates from manual observers ($N = 3$) for a subset of images. For the tibialis anterior, 100 equidistantly-spaced images in the sequence (*i.e.,* every 6th image) were selected. For the medial gastrocnemius, 84 equidistantly-spaced images in the sequence (*i.e.,* every 2nd image) were selected. Custom MATLAB code was used to manually track selected images in sequential order. Manual observers had several years of training and experience in ultrasonography and manual tracking. Agreement with manual tracking was quantified using the root-mean-square deviation (RMSD) between algorithm and manual estimates. Processing time was also compared between algorithms, and a sensitivity analysis was performed on the effect of the unknown process noise covariance parameter $c$. We used default values for all algorithm parameters, except for a process noise covariance parameter $c$ of 0.001.

### Statistics

We evaluated fascicle tracking noise and drift for algorithm estimates from the 0-20% MVC sinusoidal trial using cycle-to-cycle variability and cumulative deviation measures, respectively. These measures were statistically compared between algorithms using paired t-tests with Holm-Bonferroni corrections (*Holm, 1979*) for comparing multiple algorithms.

Linear mixed-effects regression models were used to test effects of sequence duration, image-to-image dissimilarity, and image quality, using algorithm type as a fixed effect and participants as a random effect to account for between-participant differences. The effect of sequence duration was tested by comparing overall variability from the 10th contraction cycle to overall variability from the final contraction of the sinusoidal trials. The effect of image-to-image dissimilarity was tested by comparing the change in overall variability with (1) a larger torque range during sinusoidal trials, (2) faster ramp rates during symmetric ramp-and-hold trials, and (3) faster rotation rates during passive trials. The effect of image quality was tested by comparing overall variability between low- and high-image quality sequences of the ramp-and-hold trials. These effects were tested by including fixed effects for cycle number and torque range (sinusoidal trials), ramp rate and image quality (ramp-and-hold trials), and rotation rate (passive trials) and their interaction with algorithm type in the linear mixed-effects regression models. These regressions were performed with MATLAB's *fitlme* function, using default settings (including Cholesky parameterization and maximum likelihood estimation) and parameters. Results are provided as mean ± standard deviation (sd), with sd referring to between-participants variability unless stated otherwise.

## RESULTS

Participants produced an average ankle plantar flexion MVC torque of 114.5 ± 47.7 N·m at 0 deg plantar flexion, and produced mean torques similar to the mean target torques (Table S1) in the sinusoidal trials and ramp-and-hold trials (Fig. 3). Muscle activity changed cyclically over time (Fig. S1), similarly to torque (Table S1). As expected during the sinusoidal trials, UltraTrack's estimates drifted, TimTrack's estimates were noisy, while UltraTimTrack's estimates were insensitive to both drift and noise (Fig. 3A). Similar results were obtained from the ramp-and-hold (Fig. 3B) and passive trials (Fig. 4).

UltraTimTrack's estimates had lower or similar cycle-to-cycle variability and cumulative deviation compared with estimates of TimTrack and UltraTrack (Fig. 5). Cycle-to-cycle variability of UltraTimTrack's fascicle length and fascicle angle estimates (1.4 ± 0.4 mm and 0.6 ± 0.3 deg, respectively) was smaller ($p = 0.001$ and $p = 0.002$, paired t-tests with Holm-Bonferroni corrections) than TimTrack's (3.5 ± 1.0 mm and 1.4 ± 0.5 deg), but not different ($p = 0.082$ and $p = 0.195$) from UltraTrack's (1.1 ± 0.3 mm and 0.5 ± 0.1 deg). UltraTimTrack's cumulative deviation of fascicle length and angle estimates during 0-20% MVC sinusoidal contractions (2.1 ± 1.3 mm and 0.8 ± 0.7 deg) was smaller ($p = 0.018$ and $p = 0.028$, respectively) than UltraTrack's (67.0 ± 59.3 mm and 9.3 ± 8.6 deg), but not different ($p = 0.623$ and $p = 0.476$) from TimTrack's (1.9 ± 2.2 mm and 0.9 ± 1.0 deg). Unlike its parent algorithms, UltraTimTrack's estimates had both low cycle-to-cycle variability and low cumulative deviation, indicating insensitivity to noise and drift, respectively.

Overall variability of fascicle tracking estimates from UltraTimTrack was generally comparable to or lower than that of its parent algorithms, and less sensitive to sequence duration, image-to-image dissimilarity, and image quality (Fig. 6). For sinusoidal trials,

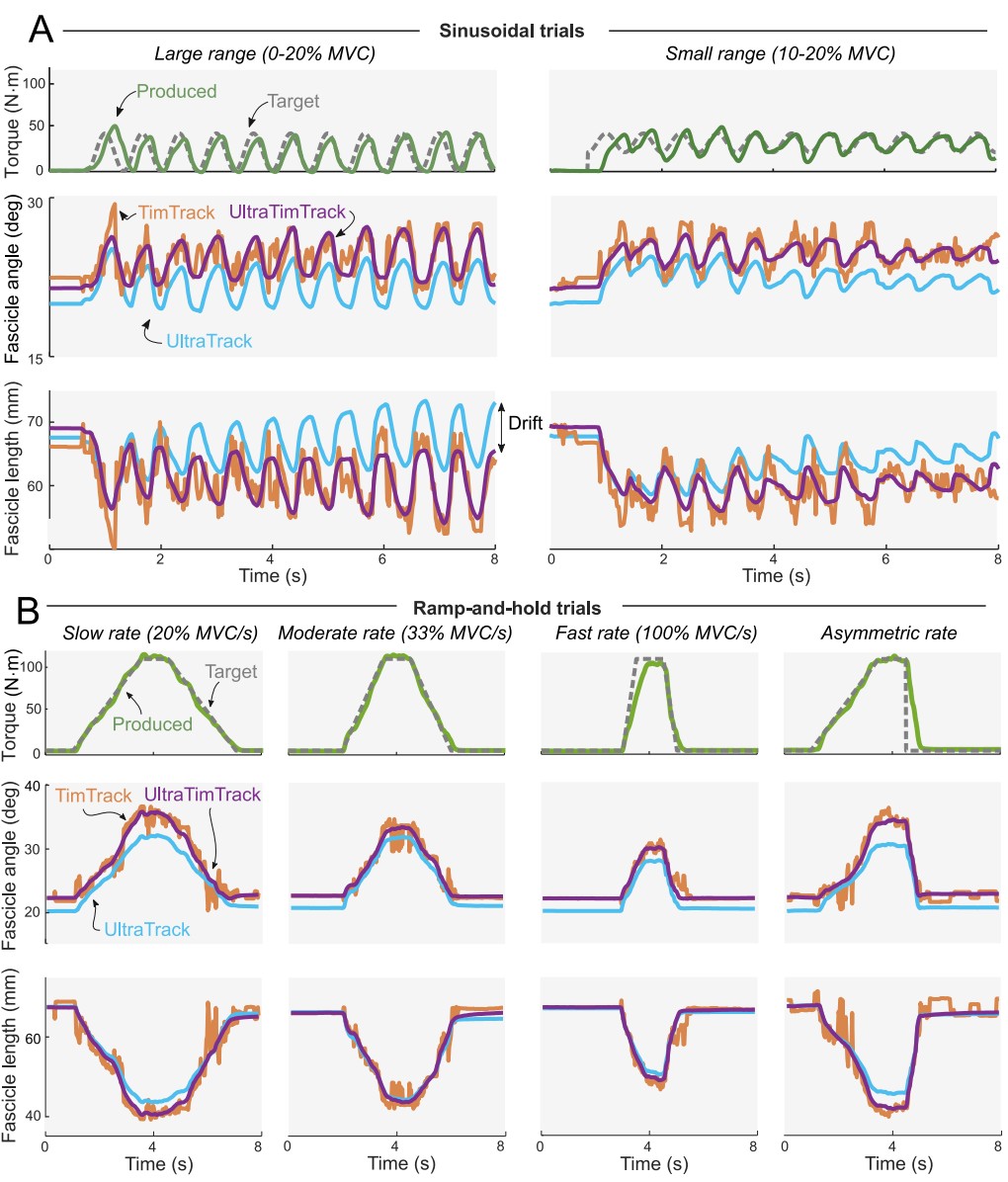

**Figure 3 Typical example of active plantar flexion torques and fascicle tracking estimates during sinusoidal trials and ramp-and-hold trials.** In both types of trials, participants produced cyclical torques at given frequencies and ranges for a duration of 80 s (first 8 s shown). Produced torque (green solid line) resembled the desired torque (grey dashed line). TimTrack (red), UltraTrack (blue) and the proposed UltraTimTrack (purple) algorithms yielded estimates of fascicle length and fascicle angle. (A) Sinusoidal trials with a large torque range (left) and a small torque range (right). UltraTrack's drift was most apparent for the fascicle length output during the large torque range trial (indicated with a double-sided arrow). TimTrack's noisiness for both fascicle tracking estimates was apparent throughout. UltraTimTrack's fascicle tracking estimates were low-noise and drift-free. (B) Ramp-and-hold trials with different ramp rates (slow, moderate, fast, asymmetric). UltraTrack's fascicle angle estimates appear offset because UltraTrack does not consider the angle of the aponeurosis; TimTrack's estimates were noisy. MVC, Maximal Voluntary Contraction.

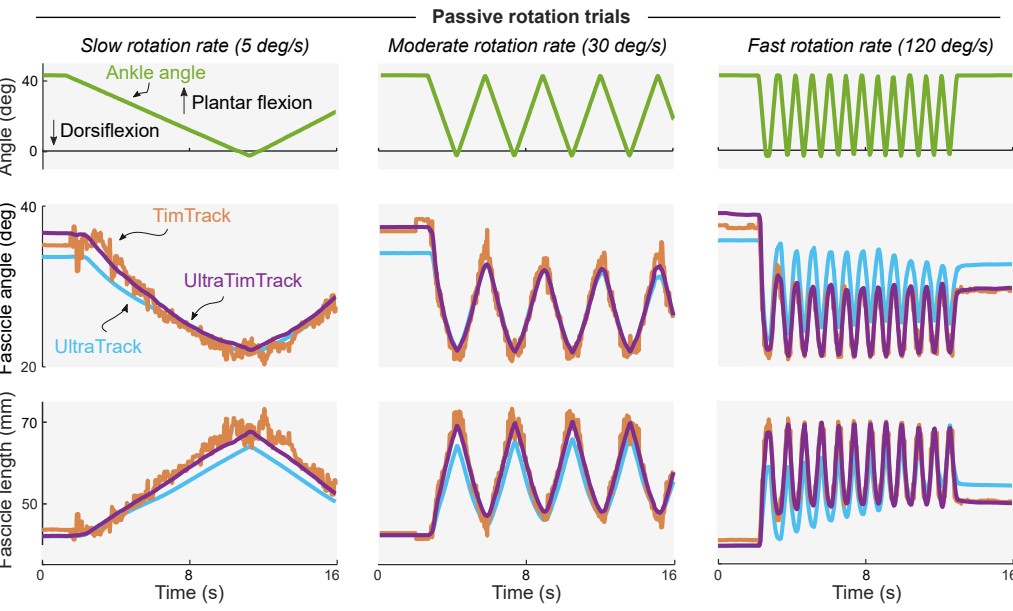

**Figure 4** **Typical example of passive plantar flexion torques and fascicle tracking estimates during passive rotation trials.** Columns show trials with different passive rotation rates (slow, moderate, and fast). Participants relaxed while the experimenter rotated their left ankle joint through its range of motion (first 16 s shown). TimTrack (red), UltraTrack (blue) and the proposed UltraTimTrack (purple) algorithms yielded estimates of fascicle length and fascicle angle. UltraTrack's fascicle angle estimates were offset because UltraTrack does not consider the angle of the aponeurosis; TimTrack's estimates were noisy.

UltraTimTrack was less sensitive to both the sequence duration and image-to-image dissimilarity than UltraTrack, as indicated by significant interaction effects between cycle number and torque range with algorithm type on fascicle length and fascicle angle variability (linear mixed-effects regression with repeated measures, Table 2). For sinusoidal trials, UltraTimTrack had lower overall variability than TimTrack as indicated by independent effects of algorithm, but there were no significant interactions with algorithm type (Table 3). For symmetric ramp-and-hold trials, UltraTimTrack was less sensitive to image-to-image dissimilarity than UltraTrack, as indicated by interactions between ramp rate and algorithm type on overall variability (Table 2). There was no significant interaction between image quality and algorithm type on overall variability, which indicates that UltraTimTrack and UltraTrack had a similar sensitivity to image quality. For ramp-and-hold trials and in comparison to TimTrack, UltraTimTrack had lower overall variability as indicated by independent effects of algorithm, but was more sensitive to both image quality for fascicle length estimates and to image-to-image dissimilarity for fascicle angle estimates (Table 3). For the asymmetric ramp trial, UltraTimTrack had a lower fascicle length and angle variability than both UltraTrack (Table 2) and TimTrack (Table 3), with no differences in sensitivity to image quality (Tables 2–3). For the passive rotation trials, UltraTimTrack was less sensitive to image-to-image dissimilarity than UltraTrack for fascicle length and angle estimates (Table 2). UltraTimTrack had lower overall variability than TimTrack for passive trials, but there was no significant difference in sensitivity to image-to-image

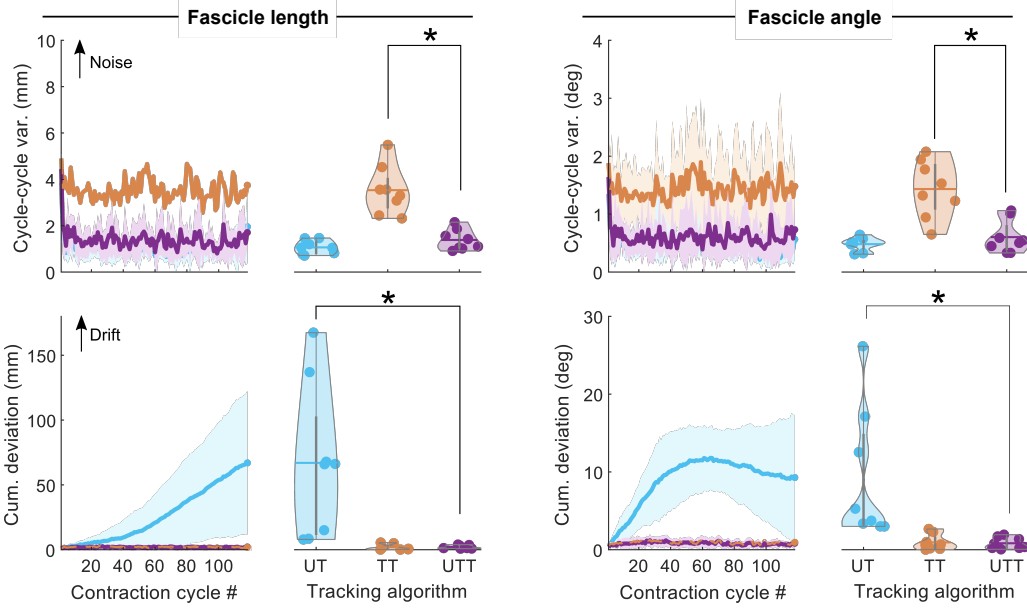

**Figure 5** Noise and drift of UltraTimTrack, UltraTrack and TimTrack fascicle tracking algorithms. Cycle-to-cycle variability and cumulative deviation of fascicle length (left) and fascicle angle (right) estimates from the proposed UltraTimTrack (UTT, purple) algorithm, and its parent algorithms: TimTrack (TT, red) and UltraTrack (UT, blue). Data are shown for sinusoidal contractions with a large torque range (0–20% MVC). Top row: Cycle-to-cycle variability (var.) for all contraction cycles (lines) and averaged over contraction cycles (violin plots). Cycle-to-cycle variability is sensitive to noise. Bottom row: Cumulative (cum.) deviation of all contraction cycles (lines) and the last cycle (violin plots). Cumulative deviation is sensitive to drift. UltraTimTrack had lower cumulative deviation than UltraTrack, and lower cycle-to-cycle variability than TimTrack; asterisks indicate significant differences ($p < 0.05$). Lines indicate mean between participants; shaded area indicates mean ± standard deviation.

dissimilarity (Table 3). Compared with UltraTrack, UltraTimTrack was less sensitive to image-to-image dissimilarity in all three types of conditions (Table 1 and Fig. S2). Overall, UltraTimTrack had similar or lower overall variability than its parent algorithms across a range of conditions, indicating robust fascicle tracking.

UltraTimTrack also compared favorably against recently-proposed HybridTrack and DL_Track fascicle tracking algorithms for the example videos that accompanied these publications (Figs. 7 and 8). Despite notable out-of-plane motion in the tibialis anterior video, both UltraTimTrack and HybridTrack yielded low-noise (*i.e.,* smooth) and drift-free fascicle tracking estimates, but with different amplitudes (Fig. 7). For the medial gastrocnemius video, both UltraTimTrack and DL_Track agreed relatively well with manual tracking, but UltraTimTrack had considerably less noise (Fig. 8). For both videos, UltraTimTrack yielded a smaller RMSD relative to manual estimates than both HybridTrack and DL_Track (Table 4). A sensitivity analysis revealed that UltraTimTrack yielded smaller RMSD's than both HybridTrack and DL_Track for a range of estimated process noise covariance parameter $c$ values spanning at least ∼4 orders of magnitude (Fig. 9). Videos of fascicle tracking by UltraTimTrack, HybridTrack and manual observers are available as Supplemental files (Videos S1–S2). UltraTimTrack's processing time was 0.2 s per image

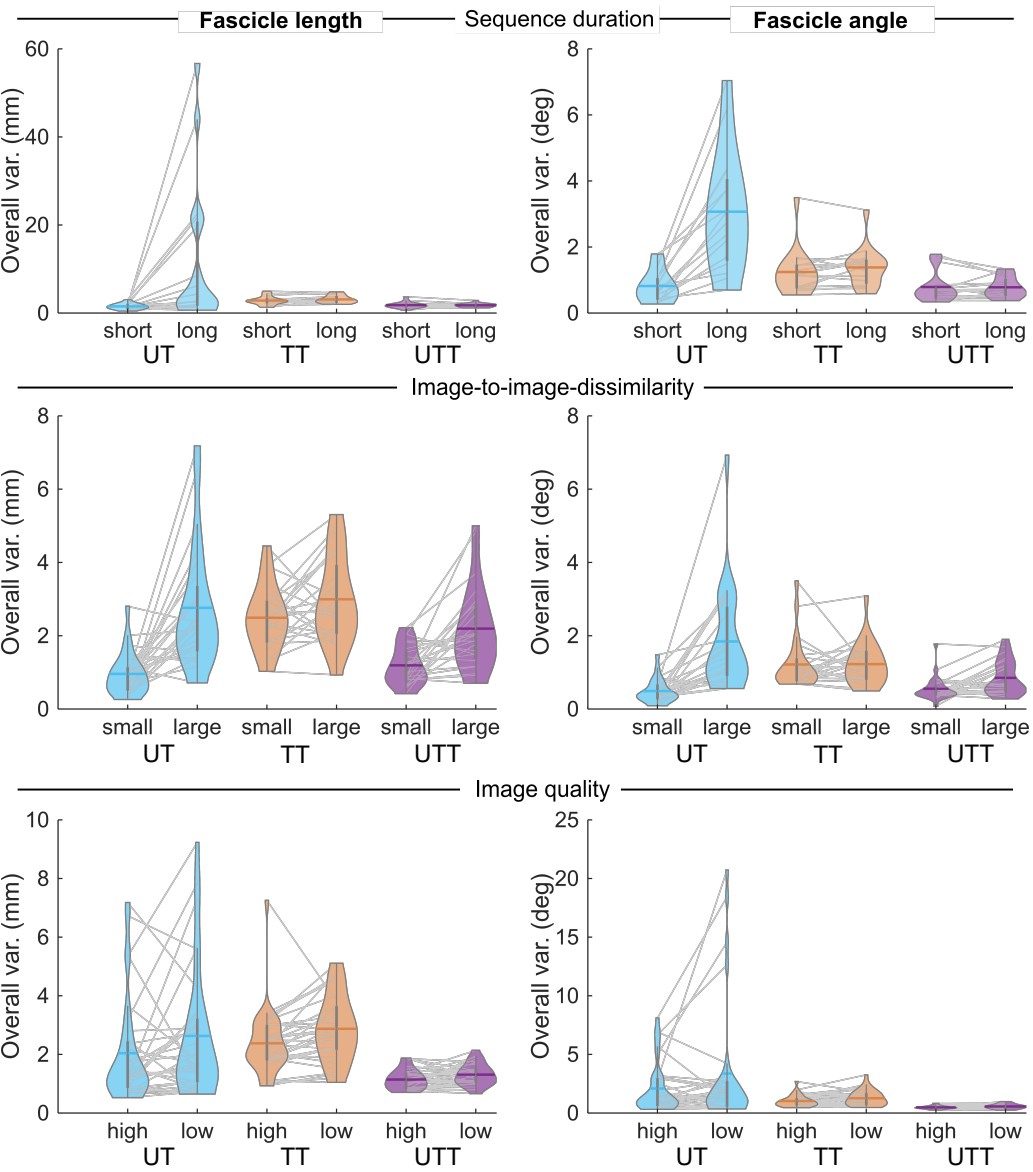

**Figure 6 Overall fascicle tracking variability of UltraTimTrack, UltraTrack and TimTrack algorithms.** Overall variability of fascicle length (left) and fascicle angle (right) estimates from the proposed Ultra-TimTrack (UTT, purple) algorithm, and its parent algorithms: TimTrack (TT, red) and UltraTrack (UT, blue). Top row: Overall variability (var.) increased with longer sequence duration. Each violin plot shows the collective estimates of both sinusoidal trials (*i.e.*, small and large torque range), either evaluated up to the 10th contraction cycle (*i.e.*, short duration) or across all 120 cycles (*i.e.*, long duration). Middle row: Overall variability increased with larger image-to-image dissimilarity. Each violin plot shows the collective estimates of one sinusoidal trial, one ramp-and-hold trial, and one passive trial, from image sequences with either small or large image-to-image dissimilarity. For sinusoidal, ramp-and-hold and passive trials, small *versus* large image-to-image dissimilarity shown here corresponds to small *versus* large torque range, slow *versus* fast ramp rate, and slow *versus* fast passive rotation rate, respectively. Bottom row: Overall variability increased with lower image quality. Each violin plot shows the collective estimates of all four ramp-and-hold trials, from image sequences with either low quality or high quality. Statistics are provided in Tables 2–3.

**Table 2 Overall estimate variability of UltraTimTrack compared with UltraTrack.** Outcomes of linear mixed-effects regression, with comparison between UltraTimTrack (UTT) and UltraTrack (UT) for fascicle length (L) and fascicle angle ($\phi$).

| Overall estimate variability | Coefficient | | Standard error | | p-value | |
|---|---|---|---|---|---|---|
| | L | $\phi$ | L | $\phi$ | L | $\phi$ |
| Sinusoidal contractions (DF = 58) | | | | | | |
| Algorithm (UT *vs.* UTT) | −5.660 | −0.652 | 3.245 | 0.366 | 0.086 | 0.080 |
| Cycle number | $-4 \cdot 10^{-4}$ | $-7 \cdot 10^{-5}$ | 0.022 | 0.003 | 0.986 | 0.976 |
| Torque range (large *vs.* small) | 0.432 | 0.127 | 2.573 | 0.290 | 0.867 | 0.663 |
| [Algorithm] × [Cycle number] | 0.095 | 0.020 | 0.032 | 0.004 | **0.004** | $\mathbf{9 \cdot 10^{-7}}$ |
| [Algorithm] × [Torque range] | 9.789 | 1.167 | 3.639 | 0.411 | **0.009** | **0.006** |
| Symmetric ramp-and-hold (DF = 90) | | | | | | |
| Algorithm (UT *vs.* UTT) | −0.113 | −0.183 | 0.339 | 0.284 | 0.739 | 0.521 |
| Image quality (high *vs.* low) | −0.156 | −0.091 | 0.236 | 0.198 | 0.511 | 0.647 |
| Ramp rate (% MVC/s) | 0.001 | −0.001 | 0.003 | 0.003 | 0.708 | 0.799 |
| [Algorithm] × [Image quality] | 0.072 | −0.001 | 0.334 | 0.280 | 0.829 | 0.997 |
| [Algorithm] × [Ramp rate] | 0.012 | 0.022 | 0.005 | 0.004 | **0.015** | $\mathbf{4 \cdot 10^{-7}}$ |
| Asymmetric ramp-and-hold (DF = 28) | | | | | | |
| Algorithm (UT *vs.* UTT) | 3.809 | 8.443 | 0.862 | 1.970 | $\mathbf{1 \cdot 10^{-4}}$ | $\mathbf{2 \cdot 10^{-4}}$ |
| Image quality (high *vs.* low) | −0.199 | −0.043 | 0.862 | 1.970 | 0.819 | 0.983 |
| [Algorithm] × [Image quality] | −1.906 | −4.79 | 1.219 | 2.787 | 0.129 | 0.097 |
| Passive rotations (DF = 44) | | | | | | |
| Algorithm (UT *vs.* UTT) | −0.609 | −0.322 | 0.333 | 0.195 | 0.075 | 0.106 |
| Rotation rate (deg/s) | 0.019 | 0.007 | 0.003 | 0.002 | $\mathbf{5 \cdot 10^{-7}}$ | **0.001** |
| [Algorithm] × [Rotation rate] | 0.010 | 0.007 | 0.005 | 0.003 | **0.044** | **0.009** |

**Notes.**
Significant *p*-values effects are shown in bold font.
DF, Degrees of freedom; MVC, Maximal Voluntary Contraction.

(Intel Core i7-10700 CPU @ 2.90 GHz), and thereby 5 times and 8 times shorter compared with HybridTrack (1.0 s per image), and DL_Track (1.7 s per image), respectively (Table 4). When parallel processing was employed (8 cores), UltraTimTrack's processing time was reduced to 0.1 per image, and thereby 10 times and 17 times shorter compared with these algorithms, respectively. UltraTimTrack thus yields better agreement with manual tracking compared with openly-available state-of-the-art algorithms, at lower computational cost.

## DISCUSSION

We proposed a Kalman-filter-based fascicle tracking algorithm that combines optical-flow-based methods with line-detection methods to improve muscle fascicle tracking in ultrasound image sequences. The proposed UltraTimTrack algorithm employs existing UltraTrack and TimTrack algorithms as modules for optical flow estimation and line detection, respectively, and combines outputs from both algorithms to reduce the time burden of fascicle tracking. Unlike existing algorithms, UltraTimTrack's fascicle length and fascicle angle estimates are low noise and drift free and are obtained at a low computational cost.

**Table 3 Overall estimate variability of UltraTimTrack compared with TimTrack.** Outcomes of linear mixed-effects regression, with comparison between UltraTimTrack (UTT) and TimTrack (TT) for fascicle length (L) and fascicle angle ($\phi$).

| Overall estimate variability | Coefficient | | Standard error | | p-value | |
|---|---|---|---|---|---|---|
| | L | $\phi$ | L | $\phi$ | L | $\phi$ |
| Sinusoidal contractions (DF = 58) | | | | | | |
| Algorithm (TT vs. UTT) | 1.147 | 0.593 | 0.294 | 0.157 | **$3 \cdot 10^{-4}$** | **$4 \cdot 10^{-4}$** |
| Cycle number | $-4 \cdot 10^{-4}$ | $-7 \cdot 10^{-5}$ | 0.002 | 0.001 | 0.844 | 0.945 |
| Torque range (large vs. small) | 0.432 | 0.127 | 0.233 | 0.125 | 0.069 | 0.313 |
| [Algorithm] × [Cycle number] | 0.003 | 0.001 | 0.003 | 0.002 | 0.363 | 0.409 |
| [Algorithm] × [Torque range] | −0.243 | −0.290 | 0.330 | 0.177 | 0.464 | 0.105 |
| Symmetric ramp-and-hold (DF = 90) | | | | | | |
| Algorithm (TT vs. UTT) | 1.782 | 0.940 | 0.215 | 0.126 | **$1 \cdot 10^{-12}$** | **$5 \cdot 10^{-11}$** |
| Image quality (high vs. low) | −0.156 | −0.091 | 0.150 | 0.088 | 0.301 | 0.302 |
| Ramp rate (% MVC/s) | 0.001 | −0.001 | 0.002 | 0.001 | 0.555 | 0.566 |
| [Algorithm] × [Image quality] | −0.453 | −0.213 | 0.212 | 0.124 | **0.035** | 0.089 |
| [Algorithm] × [Ramp rate] | −0.005 | −0.005 | 0.003 | 0.002 | 0.097 | **0.009** |
| Asymmetric ramp-and-hold (DF = 28) | | | | | | |
| Algorithm (TT vs. UTT) | 1.679 | 0.717 | 0.458 | 0.167 | **0.001** | **$2 \cdot 10^{-4}$** |
| Image quality (high vs. low) | −0.199 | −0.043 | 0.458 | 0.167 | 0.667 | 0.799 |
| [Algorithm] × [Image quality] | 0.051 | −0.009 | 0.648 | 0.236 | 0.938 | 0.970 |
| Passive rotations (DF = 44) | | | | | | |
| Algorithm (TT vs. UTT) | 1.412 | 0.804 | 0.206 | 0.155 | **$2 \cdot 10^{-8}$** | **$5 \cdot 10^{-6}$** |
| Rotation rate (deg/s) | 0.019 | 0.007 | 0.002 | 0.002 | **$3 \cdot 10^{-12}$** | **$6 \cdot 10^{-5}$** |
| [Algorithm] × [Rotation rate] | −0.006 | −0.002 | 0.003 | 0.002 | 0.055 | 0.327 |

**Notes.**
Significant p-values effects are shown in bold font.
DF, Degrees of freedom; MVC, Maximal Voluntary Contraction.

The proposed fascicle tracking algorithm yielded estimates without the drift of UltraTrack and without the noise of TimTrack. Less noise and drift were observed for time series from a representative participant (Figs. 3–4) and from participant-average summary measures (Figs. 5–6). Unlike TimTrack, UltraTimTrack's Kalman filter exploits information from optical flow to reduce noise. This allows UltraTimTrack's estimates to be smooth even for data from a single contraction. UltraTrack also uses this optical flow information, but without an automatic update step to correct drift. UltraTrack's estimates therefore drift with each contraction, causing its contraction average to be offset compared with drift-free algorithms (Fig. 3). Averaged over participants, UltraTimTrack had less drift than UltraTrack and less noise than TimTrack (Fig. 5). Less noise and drift resulted in similar or lower overall variability of UltraTimTrack's estimates across contraction cycles compared with its parent algorithms (Fig. 6). Furthermore, the proposed algorithm was generally less sensitive to factors that are known to affect tracking performance, including sequence duration, image-to-image dissimilarity, and image quality. UltraTimTrack thus provides robust estimates of fascicle length and angle changes during movement compared with existing algorithms.

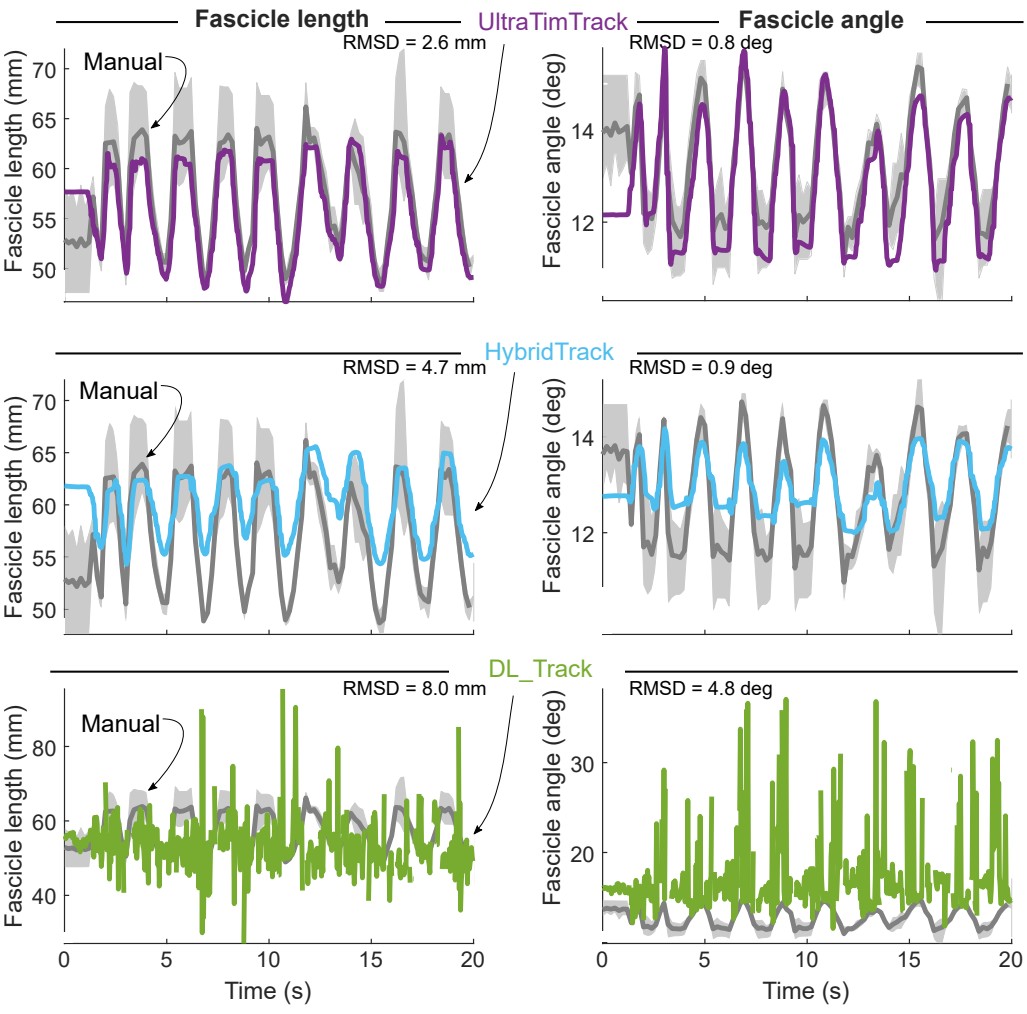

**Figure 7** **UltraTimTrack and two recently proposed fascicle tracking algorithms compared with manual tracking estimates for a tibialis anterior image sequence.** A published ultrasound image sequence from the human tibialis anterior muscle during fixed-end dorsiflexion contractions (*Verheul & Yeo, 2023*) was tracked with the proposed UltraTimTrack algorithm (purple, top row), recently proposed Hybrid-Track (blue, middle row) and DL_Track algorithms (green, bottom row), and manually by three independent observers (grey, all rows). Fascicle length estimates (left column) and fascicle angle estimates (right column) were compared between algorithms and the mean outputs from manual tracking, using the root-mean-square deviation (RMSD). HybridTrack estimates were obtained from the associated publication that included this image sequence. UltraTimTrack had a smaller RMSD from manual tracking estimates than either previously published tracking algorithm. Manual tracking outputs are shown as mean (thick grey lines) and range (light grey shaded areas) across observers. Note that the manual tracking outputs are the same for each algorithm comparison but appear different because of different scaling of the vertical axis to accommodate each algorithm's outputs.

The proposed fascicle tracking algorithm also performed well compared with a recently-proposed fascicle tracking algorithm that also combined optical flow and line detection procedures. Like UltraTimTrack, a recently-proposed hybrid method (*Verheul & Yeo, 2023*), here referred to as HybridTrack, yielded low-noise and drift-free estimates, albeit

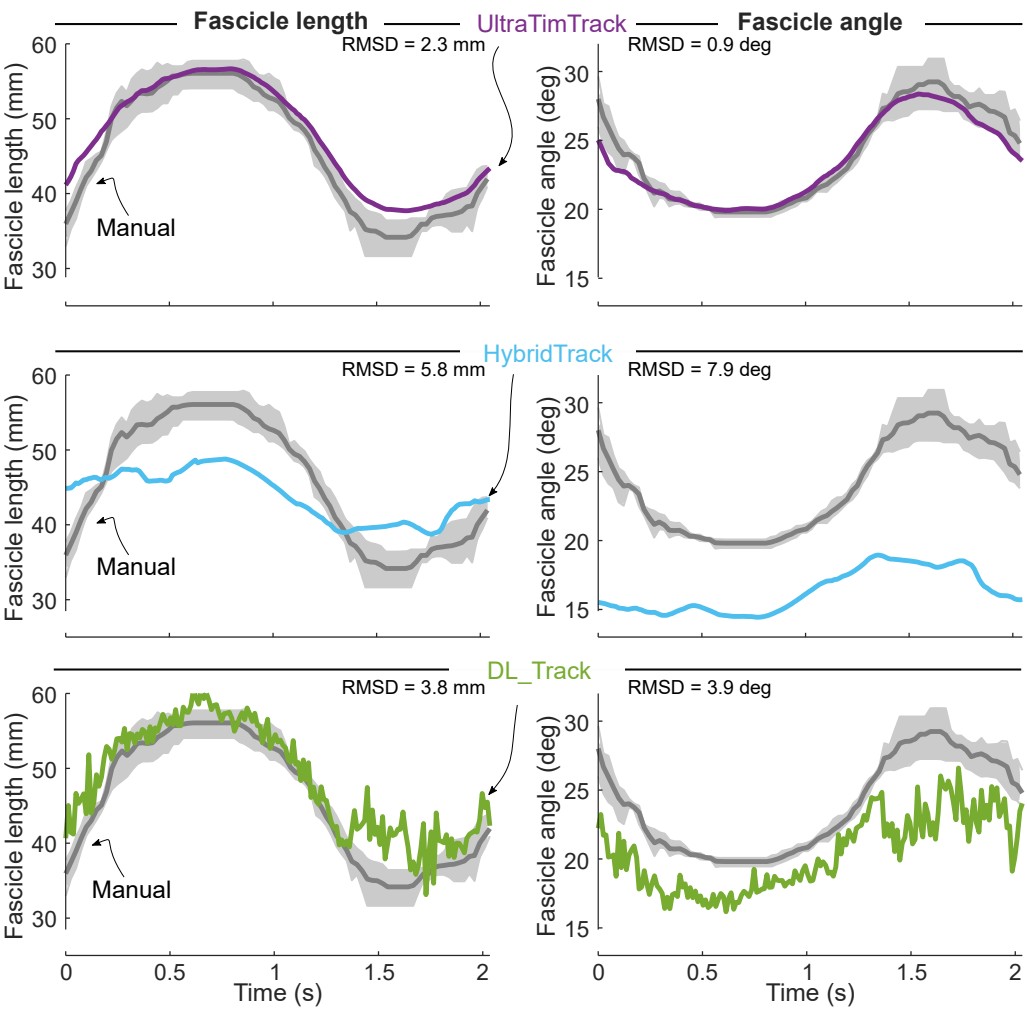

**Figure 8  UltraTimTrack and two recently proposed fascicle tracking algorithms compared with manual tracking estimates for a medial gastrocnemius image sequence.** A published ultrasound image sequence from the medial gastrocnemius muscle during "calf raise" contractions (*Ritsche et al., 2024*) was tracked with the proposed UltraTimTrack algorithm (purple, top row), recently proposed HybridTrack (blue, middle row) and DL_Track algorithms (green, bottom row), and manually by three independent observers (grey, all rows). Fascicle length estimates (left column) and fascicle angle estimates (right column) were compared between algorithms and the mean outputs from manual tracking, using the root-mean-square deviation (RMSD). UltraTimTrack had a smaller RMSD from manual tracking estimates than either previously-published tracking algorithm. Manual tracking outputs are shown as mean (thick grey lines) and range (light grey shaded areas) across observers.

with smaller amplitudes of fascicle length and angle changes (Figs. 7–8). We believe that HybridTrack yields smaller amplitudes because it (1) filters the Hough angle quite heavily, and (2) it uses the similarity transform instead of the affine transform. Its filtering of Hough angles includes a smoothing spline curve, and a 10-point moving average. Employing a similarity transform instead of an affine transform may yield smaller fascicle displacements because shear is neglected, which can be an important contributor to fascicle length and

**Table 4  Algorithm root-mean-square deviations from manual tracking and processing times.** Root-mean-square deviations (RMSDs) of Ultra-TimTrack, HybridTrack and DL_Track algorithm estimates from the mean of manual tracking by three independent observers, and corresponding algorithm processing times. Both total processing time for the sequence, and processing time per image are shown (Intel Core i7-10700 CPU @ 2.90GHz). 'Processing time-parallel' indicates UltraTimTrack's processing time when parallel processing was enabled.

|  | UltraTimTrack | HybridTrack | DL_Track |
|---|---|---|---|
| Tibialis anterior sequence from *Verheul & Yeo (2023)* | | | |
| RMSD fascicle length | 2.6 mm | 4.7 mm | 8.0 mm |
| RMSD fascicle angle | 0.8 deg | 0.9 deg | 4.8 deg |
| Processing time–regular | 124 s | 597 s | 1046 s |
| Per image | 0.2 s | 1.0 s | 1.7 s |
| Processing time–parallel | 72 s | – | – |
| Per image | 0.1 s | – | – |
| Medial gastrocnemius sequence from *Ritsche et al. (2024)* | | | |
| RMSD fascicle length | 2.3 mm | 5.8 mm | 3.8 mm |
| RMSD fascicle angle | 0.9 deg | 7.9 deg | 3.9 deg |
| Processing time–regular | 32 s | 181 s | 293 s |
| Per image | 0.2 s | 1.1 s | 1.7 s |
| Processing time–parallel | 19 s | – | – |
| Per image | 0.1 s | – | – |

fascicle angle changes (*Finni et al., 2017*). We would like to point out that this may be an error in HybridTrack, because the accompanying publication mentions that an affine transformation was used. Compared with HybridTrack, UltraTimTrack generally yielded better agreement with manual tracking, while processing time was 5 times shorter (*i.e.,* 0.2 s *versus* 1.0 s per image, Table 4). UltraTimTrack thus improves upon both tracking accuracy and processing time compared with a state-of-the-art hybrid method.

The proposed fascicle tracking algorithm also performed well compared with a recently-proposed fascicle tracking algorithm that employed machine learning. The recently-proposed DL_Track algorithm (*Ritsche et al., 2024*) is highly automated, but its estimates are quite noisy (Figs. 7–8). DL_Track's sensitivity to noise may originate from the fact that it tracks multiple (and different) fascicles in each image. The heterogeneity between line segments of the same and different fascicles can yield a noisy estimate of the dominant fascicle angle and subsequently of fascicle length. DL_Track performed better on the image sequence of medial gastrocnemius (Fig. 8) from its accompanying publication, compared with the image sequence of tibialis anterior (Fig. 7) from the HybridTrack publication (*Verheul & Yeo, 2023*). For the latter sequence, DL_Track could not detect a single fascicle or the deep aponeurosis in some images and misidentified the aponeurosis in others. Its performance on this sequence may be expected to improve when optimizing some built-in parameters and when re-trained on similar data, but this would require (1) those data being available, and (2) manual labelling. In contrast to DL_Track, UltraTimTrack showed good agreement with manual tracking in both image sequences (Table 4), despite using the same set of parameters. Furthermore, UltraTimTrack's processing time was eight times shorter than that of DL_Track (*i.e.,* 0.2 s *versus* 1.7 s per image, Table 4). Overall, UltraTimTrack

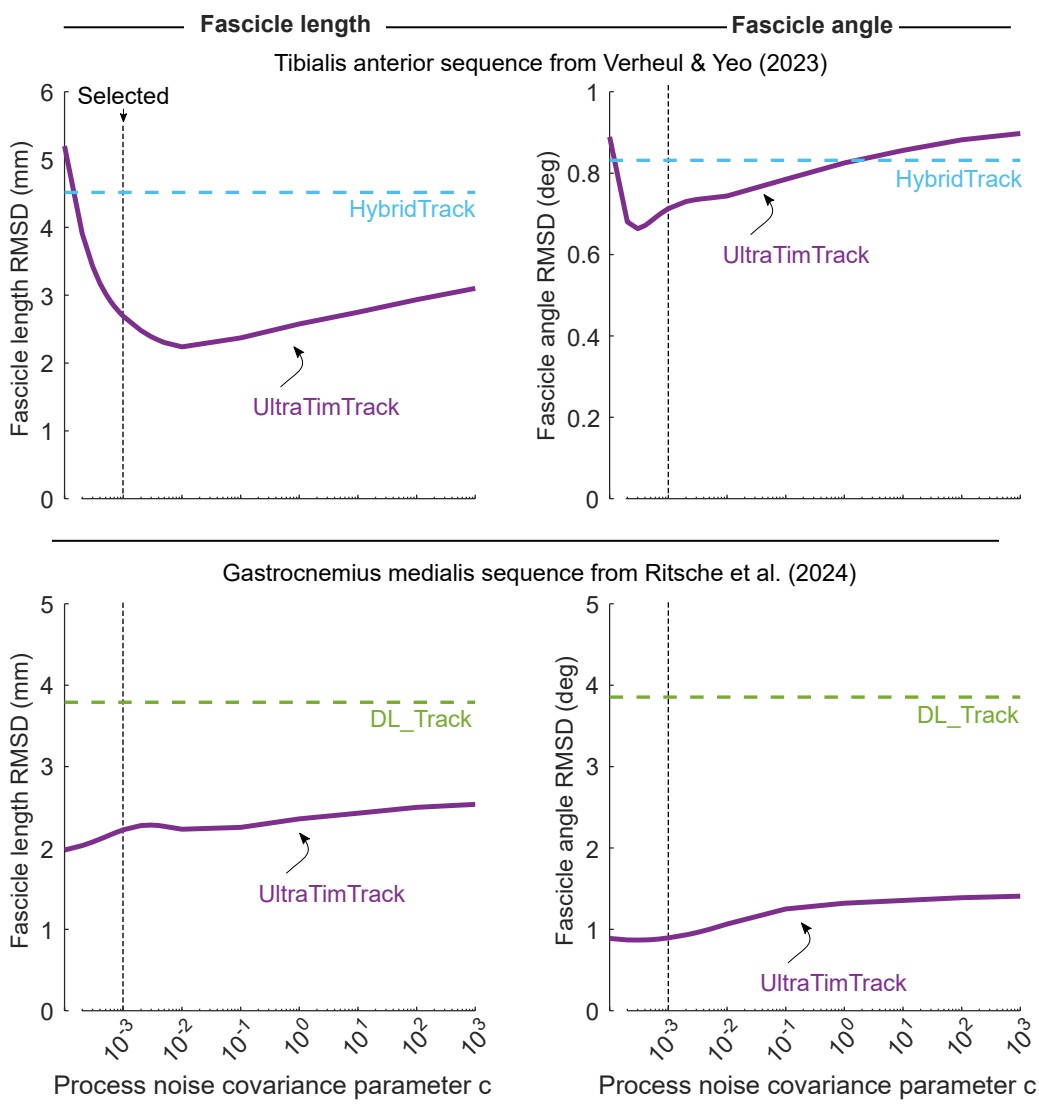

**Figure 9** **Sensitivity analysis on the effect of the process noise covariance parameter on tracking accuracy.** Process noise covariance parameter $c$ was varied, while all other parameter values were kept constant. For a range of $c$ values spanning 7 orders of magnitude, root-mean-square deviation (RMSD) of fascicle length (left) and fascicle angle (right) from manual tracking changed less than 3-fold. Top row: Fascicle length and fascicle angle RMSD of UltraTimTrack (purple solid line) and HybridTrack (blue dashed line) for the tibialis anterior ultrasound image sequence from *Verheul & Yeo (2023)*. UltraTimTrack yielded lower RMSD for a broad range of $c$ values, including the selected value (black dashed line). Bottom row: Fascicle length and fascicle angle RMSD of UltraTimTrack (purple solid line) and DL_Track (green dashed line) for the medial gastrocnemius "calf raise" ultrasound image sequence from *Ritsche et al. (2024)*. UltraTimTrack yielded lower RMSD for a broad range of $c$ values, including the selected value (black dashed line).

appears to have clear advantages compared with a state-of-the-art AI-based algorithm in terms of both fascicle tracking accuracy and computational cost.

Ultrasound-based estimates of muscle architectural changes have many potential applications in muscle physiology and movement science fields. For example,

musculoskeletal simulations (*Falisse et al., 2019*; *D'Hondt, De Groote & Afschrift, 2024*) may benefit from empirical estimates of muscle architectural changes, which could inform model parameters (*Delabastita et al., 2020*; *van der Zee, Wong & Kuo, 2024*) or improve model predictions (*Dick, Biewener & Wakeling, 2017*; *Zhang et al., 2022*). On the other hand, ultrasound tracking may also benefit from insights obtained through musculoskeletal simulations. For example, future versions of Kalman-filter-based ultrasound tracking algorithms may leverage insights into the dynamics of muscle force development derived from musculoskeletal simulations (*van der Zee, Wong & Kuo, 2024*) to predict muscle architectural changes from measurements of muscle excitation. Such algorithms could complement existing machine learning algorithms that predict muscle excitations, joint angles and joint torques from ultrasound data (*Cunningham & Loram, 2020*). Considering the low computational cost (*i.e.*, 0.1–0.2 s per image) and high apparent accuracy of the proposed algorithm (Table 4), ultrasound-derived estimates may also be used as real-time feedback to help control movements, either by a user or an external controller. The feasibility of such a 'muscle-in-the-loop' approach was previously shown for a machine-learning algorithm (*Rosa et al., 2021*). Altogether, ultrasound-based computational methods appear key to understanding the relation between muscle architectural changes and the biomechanics of human movement.

UltraTimTrack has limitations that could be addressed in future updates of the algorithm. For example, the algorithm assumes that fascicles and aponeuroses are straight lines, while they may be curved in certain cases (*e.g.*, at rest and low activation levels (*Rana, Hamarneh & Wakeling, 2014*; *Heieis et al., 2023*)). Furthermore, while UltraTimTrack is less sensitive to image-to-image dissimilarity and image quality compared with UltraTrack and TimTrack, respectively, the algorithm still requires clearly identifiable fascicles and aponeuroses. It therefore remains important to capture ultrasound images with both high frame rate and high line density, to allow both small image-to-image dissimilarity and high image quality. If users need to choose between high frame rate and high line density, this decision should be based on the anticipated rate of change in fascicle length and fascicle angle. For image sequences with poor contrast, high levels of blur, and large image-to-image dissimilarities, manual tracking may be necessary. Another limitation is that both the process noise covariance parameter $c$ and superficial aponeurosis measurement noise covariance parameter are unknown. While UltraTimTrack yielded good agreement with manual tracking for a range of parameter $c$ values (Fig. 9), it may be better to inform parameter $c$ based on independent data rather than through trial-and-error. Furthermore, we assumed a constant measurement noise covariance, even though it may be expected to vary between images. In future versions of the algorithm, measurement noise may be estimated for each image individually (*e.g.*, using the variance of a certain number of Hough angles).

## CONCLUSION

We developed a Kalman-filter-based fascicle tracking algorithm that combines existing optical-flow-based and line-detection-based algorithms to yield low-noise and drift-free

estimates of fascicle length and fascicle angle from B-mode ultrasound image sequences. The proposed UltraTimTrack algorithm has a low computational cost compared with state-of-the-art algorithms, and may be adapted for real-time fascicle tracking. UltraTimTrack is openly available to facilitate its continuous improvement, and to allow users to utilize its benefits and make modifications to address their specific needs.

### Funding
This work was supported by the Deutscher Akademischer Austauschdienst (No. 57588366) and the Open Access Publication Funds of the Ruhr University Bochum. The funders had no role in study design, data collection and analysis, decision to publish, or preparation of the manuscript.

### Grant Disclosures
The following grant information was disclosed by the authors:
The Deutscher Akademischer Austauschdienst: 57588366.
The Open Access Publication Funds of the Ruhr University Bochum.

### Competing Interests
The authors declare there are no competing interests.

### Author Contributions
- Tim J. van der Zee conceived and designed the experiments, performed the experiments, analyzed the data, performed the computation work, prepared figures and/or tables, authored or reviewed drafts of the article, and approved the final draft.
- Paolo Tecchio conceived and designed the experiments, analyzed the data, performed the computation work, authored or reviewed drafts of the article, and approved the final draft.
- Daniel Hahn conceived and designed the experiments, authored or reviewed drafts of the article, and approved the final draft.
- Brent J. Raiteri conceived and designed the experiments, performed the experiments, analyzed the data, performed the computation work, authored or reviewed drafts of the article, and approved the final draft.

### Ethics
The following information was supplied relating to ethical approvals (*i.e.*, approving body and any reference numbers):
   Ethics Committee of the Faculty of Sport Science at Ruhr University Bochum (reference: EKS V 19/2022).

### Data Availability
   The full dataset including ultrasound sequences, ankle torques, ankle angles, and ankle muscle electromyography is available at figshare: Van der Zee, Tim J; Raiteri, Brent J;

Tecchio, Paolo; Hahn, Daniel (2024). Gastrocnemius medialis muscle ultrasonography dataset. figshare. Dataset. https://doi.org/10.6084/m9.figshare.26502868.v1.

The code to reproduce the manuscript's results, figures, and statistics is available in the Supplementary File and online at https://github.com/timvanderzee/RUB_ultrasound_study.

The UltraTimTrack algorithm source code is available in the Supplementary File and online at https://github.com/timvanderzee/UltraTimTrack.

## Supplemental Information

Supplemental information for this article can be found online at http://dx.doi.org/10.7717/peerj-cs.2636#supplemental-information.

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
