# Peer review of "UltraTimTrack: a Kalman-filter-based algorithm to track muscle fascicles in ultrasound image sequences"

_PeerJ Computer Science, doi:10.7717/peerj-cs.2636_

## Round 0.1 · original submission · Major Revisions

Dear authors,
You are advised to critically respond to all comments point by point when preparing an updated version of the manuscript and while preparing for the rebuttal letter. Please address all comments/suggestions provided by reviewers, considering that these should be added to the new version of the manuscript.

Kind regards,
PCoelho

Reviewer 1 ·

Basic reporting

The authors present an improved version of TimTrack called UltraTimTrack. Tracking algorithms used on ultrasound data broadly either exploit (e.g. optical flow) or discard (e.g. deep learning) temporal information. The authors combine these two types of algorithms using a Kalman filter, and use previously established algorithms in each category as “parent” algorithms to create UltraTimTrack.

The introduction is well-written, and sufficient context, including literature references, are provided. My only suggestion would be to add a reference in line 73, after “estimates to drift away from their original values over many frames.” The suggested reference is Magana-Salgado et al., 2023 (https://pubmed.ncbi.nlm.nih.gov/37226240/). This manuscript quantifies the amount of drift in optical flow algorithms. Disclosure - I am an author on this manuscript, and please add this reference only if you agree with my suggestion.

The manuscript is structured well, and the figures are informative. My comments related to figures are:

1. Separate comparisons are made to “parent” algorithms (UltraTrack and TimTrack) and other fascicle tracking algorithms (HybridTrack and DL_Track). However, the comparison to parent algorithms is more comprehensive than the comparison to HybridTrack and DL_Track. I understand that the narrative of the manuscript might benefit from separating these ideas into two figures, but still, I can’t think of a good reason why the same set of analyses across various tasks (Figure 6) are not performed with HybridTrack and DL_Track.

2. Improve Figure 6 and the accompanying text. This figure summarizes the key findings of the paper, but the figure as it is currently presented is not easy to parse, and the accompanying text is quite confusing. The performance (based on sequence duration, image to image dissimilarity, and image quality) of three algorithms (TT, UT, and UTT) is compared in this figure on data collected during different tasks. I would suggest that the authors use a different figure organization that better allows the reader to contrast performance. For example, collapse the results to show the “main effects” along each dimension, such as the effect of image quality across all tasks performed, and then show interactions separately where appropriate.

3. Minor. In figure 4, please clarify what the green traces are.

Experimental design

Provide details on how the mixed-effects regression model was set up.

Validity of the findings

Statistical reporting should be improved. Please indicate degrees of freedom, statistic value, and confidence intervals when appropriate. Currently, only p-values are reported. For example (one-way ANOVA, F (df) = ___, p = ___, CI = __ , __ ).

I am not able to parse the following sentence - “UltraTimTrack had lower overall variability than TimTrack as indicated by independent effects of algorithm”. This sentence, with the exact wording, is repeated twice (lines 473 and 480), with different p-values. Please check for typos.

I was able to use the provided code on our data. The algorithm ran, and the GUI was intuitive and functional. The results were not as impressive as the ones presented in the manuscript, with several “outliers”, where estimates of fascicle length were over 10 m. An option for outlier rejection might make the UI useful to more users. Users should also have the option to change the limits of the axes in the graphs at the bottom of the UI - they are currently reset when the “play” button is pressed. If the authors are able to provide “ideal data acquisition settings” for using their algorithm, it would also benefit the users of the algorithm.

Line 212: where the proportionality constant is said to be chose through trial-and-error. It would be useful for the readers of this manuscript to understand how the results vary with the choice of c.

Additional comments

Line 494 - “UltraTimTrack had similar or lower estimate variability than its parent algorithms across a range of conditions, indicating both accurate and robust fascicle tracking.” How does lower variability indicate higher accuracy? I would imagine that a comparison to manual labeling would indicate higher accuracy.

Line 214: Typo - z instead of x.

Line 194: “UltraTrack optical flow is treated as input u”. Please check this. Is it only U or is it the whole system? Meaning, z_k+1 = UltraTrack(z_k)? Because if that is not the case, then explain your A and B matrices.

Reviewer 2 ·

Basic reporting

no comment

Experimental design

no comment

Validity of the findings

no comment

Additional comments

Please see the summarized comments in the attached file.

Annotated reviews are not available for download in order to protect the identity of reviewers who chose to remain anonymous.

Reviewer 3 ·

Basic reporting

Eq 12—Eq 16 are difficult to comprehend because there are a number of unlabeled variables. Moreover, the text is confusing: The authors state that they are expressing the positions in the horizontal position (x), but within the equations, they include the vertical component (y). There is no mention of how the vertical position is estimated. What are R and L?

Lines 423 - 424: The authors mention performing a sensitivity analysis, but I do not see the results of that analysis presented.

Experimental design

No Comment

Validity of the findings

No Comment

Additional comments

- Are the authors trying to track the same fascicle between the methods? Based on the supplement video that is not the case. We know that there is fascicle length variability within each muscle, so if the authors are tracking different fascicles across their methods, it could be impacting their results.

- When the authors are referring to fascicle angle, do they mean pennation angle (e.g., the angle of the fascicle within the muscle), or the angle of the fascicle within the image?

- For the supplemental video, can more information be provided? what are the red, blue, and gray lines? Why do the start and endpoints exceed the aponeurosis?

- Figure 6 is difficult to comprehend because of how cluttered the figure is and the small text.

- From lines 232-235, there are two 'First'

- Lines: 312-326 - were the conditions randomized, and if not, why not?

- The "experimental procedure" should be described before the experimental conditions.

---

## Round 0.2 · Minor Revisions

Dear authors,
Thanks a lot for your efforts to improve the manuscript.
Nevertheless, some concerns are still remaining that need to be addressed.
Like before, you are advised to critically respond to the remaining comments point by point when preparing a new version of the manuscript and while preparing for the rebuttal letter.

Kind regards,
PCoelho

Reviewer 1 ·

Basic reporting

I am satisfied with the edits made my the authors, and I have no further comments.

Experimental design

No comment.

Validity of the findings

No comment.

Reviewer 2 ·

Basic reporting

The reviewer appreciates great effort that the authors did in improving the manuscript, which significantly strengthens the quality of the current manuscript. Some minor comments can still be found below.
1. The file with all changes is supposed to only have a clean version with the highlights, but the current revised version is hard to follow.
2. When addressing the previous comment 6, it is not a good solution to put the materials in the supplemental file without addressing the comment. Even though the content is in the supplementary file, the comment still needs to be addressed.
3. In the previous comment 7, how could the comment of “careful proofreading and writing improvement” be addressed by a response “Correcting them at this stage is therefore outside of our control and we will correct them before the manuscript is published”? The writing needs to be improved before the consideration of “acceptance” or not.
4. Again, for the previous comment 8, the authors did not address the concern or issue there.

Experimental design

N/A

Validity of the findings

N/A

Additional comments

N/A

Reviewer 3 ·

Basic reporting

No Comment

Experimental design

No Comment

Validity of the findings

No Comment

Additional comments

The authors adequately addressed the reviews.

---

## Round 0.3 · accepted · Accept

Dear authors, we are pleased to verify that you meet the reviewer's valuable feedback to improve your research.

Thank you for considering PeerJ Computer Science and submitting your work.